# Towards the Automation of Infrared Thermography Inspections for Industrial Maintenance Applications

**DOI:** 10.3390/s22020613

**Published:** 2022-01-13

**Authors:** Pablo Venegas, Eugenio Ivorra, Mario Ortega, Idurre Sáez de Ocáriz

**Affiliations:** 1Aeronautical Technologies Centre (CTA), 01510 Miñano, Spain; idurre.saezdeocariz@cta.aero; 2Institute for Research and Innovation in Bioengineering, Polytechnic University of Valencia, 46022 Valencia, Spain; euivmar@i3b.upv.es (E.I.); mortega@i3b.upv.es (M.O.)

**Keywords:** infrared thermography, maintenance, industrial equipment, machine learning

## Abstract

The maintenance of industrial equipment extends its useful life, improves its efficiency, reduces the number of failures, and increases the safety of its use. This study proposes a methodology to develop a predictive maintenance tool based on infrared thermographic measures capable of anticipating failures in industrial equipment. The thermal response of selected equipment in normal operation and in controlled induced anomalous operation was analyzed. The characterization of these situations enabled the development of a machine learning system capable of predicting malfunctions. Different options within the available conventional machine learning techniques were analyzed, assessed, and finally selected for electronic equipment maintenance activities. This study provides advances towards the robust application of machine learning combined with infrared thermography and augmented reality for maintenance applications of industrial equipment. The predictive maintenance system finally selected enables automatic quick hand-held thermal inspections using 3D object detection and a pose estimation algorithm, making predictions with an accuracy of 94% at an inference time of 0.006 s.

## 1. Introduction

It is well known that the maintenance of industrial equipment, machines, or facilities tends to extend their useful life, ensure correct performance for a longer amount of time, improve efficiency, reduce the number of failures, and increase the safety of their use [1]. Maintenance management has evolved significantly over time. Progress has been made in the way that maintenance is applied to industrial equipment [2]. Initially, corrective maintenance was the common rule at the early beginning of factory use, where equipment failures were solved as they occur. This type of maintenance was accepted at early stages of the industry when downtime was not critical. However, after the second industrial revolution, maintenance evolved to apply preventive maintenance, where the equipment was periodically reviewed, and certain components were replaced based on statistical estimates, often provided by the manufacturer. The drawback of this concept of maintenance was the high associated costs due to strict replacement deadlines, which were often overestimated.

The maintenance evolved again up to the current tendency of applying a predictive approach, which tries to predict the failure of a component such that the component is replaced shortly before the failure occurs. Thus, equipment downtime is minimized, and component life is maximized. By applying a predictive strategy, the useful life of an asset can be extended up to five times more than if a preventive strategy is followed [3]. The predictive maintenance is based on the measurement of various key parameters of a specific industrial process that are closely related to the life cycle of the component. Thus, the remaining life of the component is reliably estimated by defining the history of the relationship between the physical magnitude and the condition of the machine. This history is produced by monitoring and taking measurements of certain magnitudes at periodic intervals until the component breaks or fails. Within predictive maintenance, there are different disciplines or techniques, such as vibration analysis, oil analysis, and ultrasound inspection, among others. Each of these techniques are based on the measurement of different magnitudes and are most suitable for the specific diagnosis of failures [4].

A relatively novel predictive maintenance technique that is gradually spreading in the industrial sector is an infrared thermography (IRT)-based technique [5]. IRT is a technology that measures radiation within the infrared spectral range that an object emits and relates it to the actual temperature of the object. Taking into account that the vast majority of failures in industrial equipment and machinery are preceded by significant temperature changes, it may be concluded that the thermal data collected with a thermal sensor is a valuable source of information and can identify problems at an early stage so that they can be corrected before becoming worse and more expensive to repair. Infrared thermography has great potential as a predictive maintenance tool for equipment and facilities that can optimize industrial maintenance costs [6,7,8,9,10]. Although the characteristics of the IRT make it applicable to multiple areas and provide important advantages over other techniques, its scope is still quite limited. Nowadays its use is mainly aimed at detecting existing faults, rather than a true predictive strategy that anticipates the occurrence of failures.

Among the most promising technologies that are currently being applied in a wide variety of applications with notable results is machine learning (ML). Machine learning is a branch of artificial intelligence (AI) that enables accuracy in predicting outcomes without explicit instructions to do so. Instead, machine learning algorithms use historical data as input to predict new output values after a training process through which the ML model captures the fundamental features of the process involved in the production of outputs. ML is currently being applied widely in our lives, for example in search engines, virtual assistants, online customer support [11], traffic predictions [12], video surveillance [13], email spam and malware filtering, and online fraud detection [14].

Another emerging technology whose effectiveness in industrial maintenance has already been proven is augmented reality (AR). Among the multiple advantages of this technology in the field of equipment maintenance, the reduction of the costs associated with the acquisition of operation and repair skills, as well as the reduction of risks associated with technical personnel by guiding the execution of maintenance tasks more accurately, is worth mentioning. AR has received a growing amount of attention by researchers in the manufacturing technology community, because AR can be applied to address a wide range of problems throughout the assembly phase in the lifecycle of a product, e.g., planning, design, ergonomics assessment, operation guidance, and training [15]. Numerous studies show promising results in enhancing human performance using AR equipment in this area [16,17]. Specifically, the use of AR reduces the number of mistakes, enhances security, and accelerates the task. Additionally, AR-assisted maintenance tools also lead to a reduction in the cognitive load of workers [18].

The application of ML in the field of maintenance is also being investigated with the aim of producing more accurate predictions of failure and reducing downtime as much as possible. Specifically, several studies have been conducted to apply ML with IRT to automate the detection of malfunctions in different types of equipment. The results obtained in these studies are promising, but certain limitations were encountered, so further research is required to reach a robust application of ML with IRT in industrial environments. In [6], the authors applied a Support Vector Machine (SVM) model to infer malfunctions in electric connections, obtaining outstanding results compared to conventional thresholding methods. Despite the high accuracy obtained, in this study, the classification was conducted pixel-wise in the thermal image, so the procedure is expected to be quite time-consuming, preventing its application in real-time conditions. The classification procedure is generally carried out into two categories, correct and defective [6,8,19], in up to three cases, including non-faulty hotspots [20]. Some authors use the values of temperature increment (ΔT) as input features, which is a common standard [21,22], while others use such features as maximum and minimum temperatures [23], RGB color scales [24], texture features [25], and Zernike moments [26]. In [8], the authors used discriminant analysis to select optimal features from a wide set of statistical parameters and finally selected 10 features. However, the regions of interest used to compute values for training the model and making predictions was conducted manually, precluding the automatic application of the methodology.

The maintenance cost can reach up to 60–70% of the total production cost during the production lifecycle according to the research in [27], so it is an important cost to optimize. It is reasonable to assume that the combination of AR, IRT, and ML would bring even more benefits, as stated in the work of [28]. The present study tries to advance toward the automation of infrared thermography inspections for industrial maintenance applications. Unlike other previous related studies, this study is focused on the development of a methodology to produce accurate predictions in industrial maintenance activities carried out with infrared thermography to be implemented in portable platforms, requiring reduced computational cost, being able to classify numerous different states corresponding to specific failures and malfunction situations, and easily scalable to other industrial equipment. For this purpose, the recent AR system called MANTRA [28] has been enhanced with a machine learning module for computing thermographic information automatically and robustly and predicting possible failures before they occur. This functionality has been added to the previous ones for automatic task guidance, component location, and specific temperature measurement.

The results of this work provide advances towards the robust application of ML combined with IRT to maintenance applications of industrial equipment. A new module for the MANTRA system that informs of failures or abnormal behaviors of industrial equipment analyzing thermal images is developed. Moreover, AR is also combined with these technologies and implemented in a portable device making the MANTRA system, which is, to our knowledge, the first AR system that has been used in combination with IRT in the field of industrial maintenance. MANTRA enables quick hand-held thermal inspections using 3D object detection and pose segmentation in order to segment IRT information at the component level. Experimental data have been produced in this study by monitoring electric industrial equipment and inducing malfunctions in a controlled way. Finally, different ML strategies have been analyzed and assessed to finally define the optimal procedure to be implemented in the MANTRA system.

The paper is organized as follows. In Section 2, the materials and methods used in the study are described, including data processing techniques, predictive models, study cases, and hardware and software equipment. The main results obtained are presented in Section 3, describing the applied procedures and showing the outcomes in the precision and performance evaluation. Section 4 is dedicated to further discussion of the results and the proposal of a general approach for predictive maintenance system development. Section 5 is devoted to the final conclusions and suggested future actions.

## 2. Materials and Methods

### 2.1. Thermographic Selected Features

Infrared thermography is a technology used in many applications with the main objective of measuring temperature remotely and without contact. From the temperature it is possible to determine the value of magnitudes related to heat exchange phenomena that take place in different physical processes. Temperature measurement by IRT consists of the measurement of the infrared radiation emitted by an object and its subsequent conversion to temperature values [5]. Infrared thermography is a technology that is currently used in numerous applications. The special non-invasive and non-contact measurement characteristics, together with the harmless health effects, make the IRT a useful instrument in areas such as medicine [29], agriculture and livestock [30] or surveillance [31], among others.

The high noise content in the data is one of the most important issues affecting IRT, which can hide the thermal effects of existing defects, making them undetected. Although this problem is being taken over by technological advances in the IR sensors, it is still a major limitation in the detection capability of the thermographic technique. Different processing algorithms have been proposed to reduce the effect of the noise in thermographic data [32,33,34,35]. These algorithms enhance the detection of defects and simultaneously reduce the noise content of the image.

On the other hand, the machine learning models used in this study must be fed with feature vectors that are representative of the operating condition of specific pieces of equipment so that they capture their fundamental characteristics and make it possible to infer the actual conditions for new vector inputs. In order to obtain features representative of the operating conditions and avoid the limitations produced by the noise content of the thermographic images, some of the most common processing algorithms used in the IRT NDT field were analyzed in this study. The processing methods synthesize the thermal behavior shown by the components of the equipment under inspection so that they contain the information necessary for determining the malfunction operating conditions.

The processing algorithms used in this study are described below.

#### 2.1.1. Thermographic Signal Reconstruction (TSR)

Thermographic signal reconstruction is a processing technique widely used in IRT NDT for filtering noise, producing an increment in the signal-to-noise ratio and enhancing the detection of defects [32,36]. The application procedure consists of several steps including the polynomial fitting, through which the temperature–time history of each pixel is fitted to an n-degree polynomial. In this study, TSR was not applied for data processing but only the polynomial fitting stage. The other steps were discarded because of the lower effectiveness when applying TSR in passive inspections.

Polynomial fitting was conducted by the expression (Equation 1), where *T* is the temperature values, *t* is the time-independent variable, and $\alpha_{i}$ are the fitting coefficients. The resulting coefficients represent the thermal behavior of the region under analysis, so they can be used as features in the training process of predictive models.
(1)$$T\left(t\right)=\alpha_{n}t^{n}+\alpha_{n-1}t^{n-1}+\cdots +\alpha_{1}t+\alpha_{0}$$

#### 2.1.2. Spectral Analysis

This processing technique applies the Fourier transform (FT) to the raw thermographic data sequence [37]. The initial thermographic sequence is transformed from the time to the frequency domain by the relation (Equation 2), where $F_{n}$ is the *n*th complex Fourier coefficient with $Re_{n}$ and $Im_{n}$ as real and imaginary parts, respectively, *T* is the temperature value of each pixel, and *N* is the total number of frames in the thermographic sequence.
(2)$$F_{n}=\sum_{k=1}^{N-1}T\left(k\right)e^{\frac{\left(2\pi ikn\right)}{N}}=Re_{n}+Im_{n}$$

#### 2.1.3. Principal Component Analysis (PCA)

Principal component analysis is a processing technique based on the second-order statistics of the initial data, usually used for data compression and the detection of linear relationships [38]. PCA is performed with thermographic data by applying the singular value decomposition of the temperature–time history of each pixel according to the following expression:(3)$$\nu_{p}=U_{S}^{T}A$$
where *U* is the calculated eigenvector matrix, *A* is the initial temperature matrix, and $\nu_{p}$ is the result considering the first *s* principal components, where the number of components are selected to retain the desired amount of variance in the corresponding *s* eigenvalues.

#### 2.1.4. Higher Order Statistics (HOS)

Higher-order statistical moments computed over the surface temperature distribution in thermographic NDT is a processing technique that provides statistical information of the temperature evolution in a unique image [34]. The most informative moments for thermographic applications were demonstrated to be the standardized central moments calculated by
(4)$$\mu_{n}=\frac{E[{\left(X-E[X]\right)}^{n}]}{\sigma^{n}}$$
where E[X] is the mean value of the distribution, σ is the standard deviation, and *n* is the order of the moment. The specific statistical moments used in this study to characterise the different operating conditions are the third and fourth standardized statistical moments, also known as skewness $\mu_{3}$ and kurtosis $\mu_{4}$, respectively.

### 2.2. Machine Learning Models

Machine learning is currently a well developed science that has numerous application possibilities based on different algorithms and processing techniques [39,40]. There are simple models that produce acceptable results under certain simplicity conditions, and there are complex models that capture the deep, non-linear properties, e.g., deep neural networks. The application of interest in this work, that is, the maintenance of industrial equipment, requires that the computational cost of machine learning models does not reduce the performance of the calculations carried out or limit the correct inspection activities.

This study focuses on supervised predictive models, which are models configured with known and labeled data at an initial stage known as training, which can subsequently provide predictions for new data not seen before [41]. In this way, taking into account previous information regarding the operating conditions of equipment, expected conditions may be inferred with new data. Within the supervised ML models, two types of predictions may be distinguished: predicting a label or predicting a continuous real value [40]. This study deals with labels that correspond to specific discrete operating conditions.

Among the numerous classification models currently available within the machine learning field that covers the aforementioned requirements, the following models are assessed in this work.

#### 2.2.1. Random Forest (RF)

Random forests consists of a large number of individual decision trees that operate as an ensemble [40]. A decision tree is a supervised learning technique that can be used for both classification and regression problems [42]. It is a classifier with a tree structure, where internal nodes define the features of the dataset, branches define the decision rules, and leaf nodes represent the outcomes.

A random forest model constructs a certain number of randomized decision trees during the training stage and makes predictions by averaging their individual results [43]. Each tree in the random forest makes a prediction, and the class with the highest number of predictions becomes the final result. Random forests can be applied to a wide range of prediction problems and have few parameters to tune. Despite being a relatively simple model, the method is generally recognized for its accuracy and its ability to deal with small sample sizes, high-dimensional feature spaces, and complex data structures.

#### 2.2.2. K-Nearest Neighbors (KNN)

The nearest neighbors method is non-parametric, and its principle of operation is to find a previously defined number of training samples closest in distance to the new point and to predict the corresponding new label according to them [44]. It does not construct a general behavior model; it simply stores instances of the training data. Classification is then performed from a simple majority of the nearest neighbors of each point, so the data class with the most representatives within the nearest neighbors is assigned to the new point.

Nearest neighbors methods are known as non-generalizing ML methods, since they simply compare the new data to the data already known. The number of samples can be constant (k-nearest neighbor learning) or vary based on the local density of points (radius-based neighbor learning). Standard Euclidean distance is the most common choice for the distance, but any metric measure can normally be used.

They are often accurate in classification situations where the decision boundary is very irregular, and despite its simplicity, nearest neighbors have been successfully applied in a large number of classification and regression problems, including handwriting recognition and satellite imagery [45,46].

#### 2.2.3. Support Vector Machines (SVM)

Support vector machines are supervised learning methods that can be used for classification, regression, and outlier detection [47]. These methods construct a hyperplane or a set of hyperplanes in a high or infinite dimensional space, which is used for computation. Convenient separation is achieved by the hyperplane that has the largest distance to the nearest training data point of any class, since in general the larger the margin, the lower the generalization error of the classifier.

SVM is more effective in high-dimensional spaces and provides satisfactory performance with a clear margin of separation between classes. An important quality of this method is its high effectiveness in situations where the number of dimensions is greater than the number of samples available. Applications where SVM has been successfully used include the classification of genes and patients on the basis of genes and other biological problems [48] as well as handwritten character recognition [49].

#### 2.2.4. Artificial Neural Networks (ANN)

Artificial neural networks is a specific type of ML model that is composed of many interconnected computational units, called neurons [50]. Although each individual neuron provides limited learning capabilities, when many neurons work together interconnected, their combined effect shows a significant learning performance.

ANNs are models inspired by the brain, where neurons are simple representations of their biologic equivalents. The inputs to an artificial neuron correspond to raw data or outputs from previous neurons. The transfer function sums all these inputs together, and, if the result reaches a specific limit value, the activation function generates an output signal. This output signal goes to a final output or other neurons depending on the ANN architecture. There is a long list of neural network applications currently in use in different industries, such as high-performance self-driving [51], the prediction of code sequences [52], voice synthesis [53], and vision systems [54].

### 2.3. Study Cases

There is a wide variety of industrial equipment to which maintenance activities are applied on a regular basis, such as engines, rotary systems, heat exchangers, and refrigerators. To limit the scope of the study and define an affordable scope, this work is focused on the maintenance of electronic equipment that works through cycles of transitory operation. Equipment operating continuously, for which previous studies have reported remarkable results, have not been analyzed. In this work, electronic systems that remain in a paused state until they perform their operation after being activated either manually, remotely, or even programmatically have been analyzed.

To carry out this study, an electronic board was used as an example of the type of devices of interest. This board is a euro 230m2 model (VDS Automazioni srl, Spoltore, Italy) and is normally used to control the automatic opening and closing of swing doors. This device is directly connected to a 220 V voltage and is in charge of powering two direct current electric motors, which produce the necessary force to make one or more actuators move the elements that open and close the gate (Figure 1a).

In order to carry out the experimental study on this board in a controlled way, two variable resistors of 200 Ω (Vishay Intertechnology, Inc., Malvern, PA, USA) have each been connected to the output, instead of the actuation system consisting of a motor and an actuator. (Figure 1b). By means of a clamp that can be adjusted in any position along the resistance, it is possible to control the effective value of resistance in the output of the board. Based on this laboratory setup, which reproduces the actual operation of the board, a series of failure cases was defined and analyzed to develop a predictive model capable of accurately classifying them. Some of these failures disable the operation of the electronic device, while other cases suppose abnormal operating conditions that will lead to the total failure of the device in a short period of time. The failure cases that have been analyzed are the following:Bad contact in the input connector. This malfunction was induced by releasing the wires in the input connector. This type of malfunction produces local overheating that eventually causes the break of the connector and a total service interruption.Overload in the outputs to the actuators. This malfunction case corresponds to irregular operating conditions caused by abnormal circumstances in the output loads. At the laboratory level, this case was simulated by reducing the resistance connected to the board, therefore increasing the operating current flow. In a real scene, these circumstances could correspond to disconnections in a gate system, for instance. Eventually, these malfunctions will cause the break of the electric motors commanded by the board.Underload in the outputs to the actuators. This malfunction case is the situation opposite to the previous overload case. At the laboratory level, this case was simulated by increasing the resistance connected to the board, so mechanical misalignment or excessive friction between moving elements in the real situation is simulated.Failure in the fuses. This case is produced by the break of a fuse and, contrary to the previous cases, requires corrective rather than predictive actions. This failure prevents the correct operation of the device and disables the output current. There are two fuses in the board, so the failure is produced in a different output, depending on the affected fuse.Failure in the relay. This case is produced by the malfunction of a relay, which was induced at the laboratory level by avoiding the relay contacts to join correctly. This case also requires corrective actions, and the apparent effect to the system operating conditions is the same as in the fuse failure case, disabling the output current.Reference state. The case corresponding to the correct operation was considered as the reference state. In this situation, the board outputs a direct current to two resistors of 90 Ω for a period of 7 s after activation of the board.

### 2.4. Hardware and Software Equipment

The MANTRA system consists of two cameras (RGB-D and infrared) mounted and aligned on a portable device that processes the information and displays it to the user. The infrared (IR) camera used in this analysis was a Xenics Gobi-640-GigE model (Xenics NV, Leuven Belgium) with a spatial resolution of 640 × 480 pixels that works in the 8 μm to 14 μm spectral band. The low weight (263 g) and size (49 × 49 × 79 mm${}^{3}$) of the camera make it ideal as a hand-held device for carrying out maintenance inspections of industrial equipment. The second camera employed is an IntelRealSense d415 RGB-D camera (Intel Corp., Santa Clara, CA, USA) with a color resolution of up to 1920 × 1080.

The inspections conducted by the thermographic camera were passive; i.e., no additional thermal stimulation was applied to the inspected object. However, the normal operating conditions of the board produced heat that was detected by the IR sensor. The data captured by the thermographic camera was analyzed using the processing techniques described in Section 2.1 to extract the features that enabled the classification of the cases under study. Each operating cycle of the board was completed in 5 s, but the thermal process was recorded over 7 s, so the temperature increase and decay were captured in the infrared sequence. The infrared and RGB-D cameras recorded frames at a rate of 10 Hz (Figure 2) from a distance of around 1 m from the board. At this inspection distance, the IR camera produced 1.1 mm/pixel images and the RGB-D camera produced 0.46 mm/pixel images. The dataset used to train the models and perform predictions consisted of 20 thermal sequences per study case.

The software used to process the infrared images as well as configure, train, and test the different predictive models was coded in the Python programming language. The image processing algorithms were implemented making use of the Numpy and Scipy modules, while the predictive models were developed with the Scikit-Learn, Keras, and Pandas modules.

The MANTRA system demonstrator was developed and tested on a Surface Book 2 platform (Microsoft Corp., Redmond, WA, USA) with an Intel core i7 processor, an NVIDIA Geforce GTX 1050 2 GB GPU, 8 GB RAM, and 256 GB SSD. The machine learning models, the 3D object detection and pose estimation algorithms, and the data processing techniques and applications were developed in C++ and work on Windows 10. Specifically, the main dependencies employed were OPENCV [55], Darknet [56], Open3D [57], and CUDA [58].

## 3. Results

### 3.1. Selection of Regions of Interest

The first stage in the development of the predictive model for electrical failures was to determine the components of the electronic board, whose behavior will enable the classification of the different operating conditions. Image classification technologies that process the entire image and that do not require one to determine areas from which features can be extracted are currently being developed. These technologies are part of the science of deep learning, and, in addition to finding limitations in dealing with infrared images, there is no public database of infrared images that can be employed for training in the industrial maintenance field, nor are there pretrained networks that employ transfer learning techniques. These factors, together with the necessity of powerful GPUs and long training times, makes the use of deep learning techniques difficult to recommend for a general IRT methodology in industrial maintenance.

In this study, a series of areas within the board that contain the most informative and representative points of the different operating conditions analyzed were identified. Sets of decision trees were used to identify these points. Specifically, the identification of the regions of interest was carried out by analyzing the mean impurity reduction calculated on a series of independent decision trees and by averaging the values produced by each one to reduce the variance of the results (bagging) [41,59].

Each decision tree is a set of internal nodes and leaves. In the internal nodes, the feature is selected to divide the dataset into two separate sets. In classification tasks, the features for internal nodes are selected by means of the Gini impurity or information gain [43]. Gini impurity calculates the amount of probability of a specific feature that is classified incorrectly when selected randomly, so we can measure how each feature decreases the impurity of the split. A higher mean decrease in Gini indicates higher variable importance. The average over the set of trees is the measure of the feature importance.

Each pixel in the thermal image was considered a different feature for the importance analysis. The matrices of pixel values that represent the thermal images were converted into row vectors. By repeating this process for all the images in the dataset, a matrix of size M × N was constructed, where M is the number of images in the dataset, and N is the number of pixels in each of the thermal images. Using this new matrix, the previously described classification model was built, with a total number of 1000 estimators and 128 maximum characteristics to be considered in each split, and the importance of the different features was calculated, that is, the importance of each pixel in the image. Figure 3a shows one of the results obtained from the importance analysis for the case of data obtained from the coefficient $\alpha_{0}$ of the polynomial approximation processing technique. This same analysis was carried out for all data processing techniques, obtaining similar results regarding the level of importance of the different pixels in the thermal images.

As a result of the importance analysis, eight areas of highly informative content were identified within the electronic board (Figure 3b,c). These zones were labeled from left to right and from top to bottom, and corresponded to different components and set of components. Specifically, the region A had a circular shape with a diameter of 9 mm and corresponded to a rectifier bridge; the regions B and C had rectangular shapes of 24.5 × 9 mm and corresponded to fuses; the region D had a rectangular shape of 3.5 × 1.5 mm and corresponded to an SMD resistor; the region E had a rectangular shape of 14 × 12 mm and corresponded to the power input connection; the region F had a rectangular shape of 18 × 12 mm and corresponded to the output connection to loads; and the regions G and H had rectangular shapes of 12 × 14 mm and corresponded each one to a set of two through hole resistors and a triac. From this analysis, it can be deduced that the identified components must be monitored in order to determine and classify the operating conditions of the board. Once the components were identified, the predictive models were built, and their inference capacity was analyzed. However, although the key components had been identified in a previous stage, after the final model definition, the positioning of the analysis and data extraction areas was carried out automatically through the use of 3D object detection and pose estimation techniques.

#### 6D Object Pose Estimation and IRT Image Segmentation

The 6D object pose estimation employed in the MANTRA system is LINEYOLO [28]. This algorithm employs the deep learning objection detection algorithm Yolov4 [56] for detecting the object in the image and later employs a modified LINEMOD algorithm [60] to estimate the 6D pose. Although this algorithm is quite fast, obtaining a stable, robust, and real-time 6D pose requires the employment of a 6DOF pose tracking algorithm. MANTRA employed the algorithm proposed by Tjaden et al. [61] to achieve these objectives. Essentially, this tracking method iteratively resolves a non-linear optimization problem of the parameters that define the rigid transformation of an object between two consecutive frames.

One of the main features of the MANTRA system is that it can augment the color information integrating the IRT information. In this way, it is possible to show the temperature of an isolated 3D object, as well as that of its various components, on top of the color image with associated virtual information, both precisely and in real time. To achieve this IRT fusion, it is necessary to calibrate each of the sensors, place them in the same coordinate system (extrinsic parameters) employing a stereo calibration, and overlap the registered images using the information of the previously calculated 3D object pose. The details of the IRT fusion can be found in [28].

The procedure can be seen in the schematic in Figure 4. The IRT fusion allows for a temperature segmentation and a segmentation based on the 6DOF object pose. The temperature segmentation function is employed to select a range of temperatures to show on top of the color image. This could be employed, for example, to find hot spots or trigger alarms when the temperature overpasses a certain level. Moreover, the segmentation based on the 6DOF object pose function uses this information to project the 3D model into the image and employs that mask to segment the IRT information robustly and independently of the point of view. As a result, the IRT information can be processed at the component level automatically in order to detect malfunctions or errors in a hand-held mode, overpassing the traditional problem of performing inspections with the thermal camera at a specific position and orientation. Figure 5 depicts four different points of view of the electronic board acquired with the MANTRA system, where the triac component controlling the power source is being monitored (only the IRT information of this component is shown in the image).

### 3.2. Performance Assessment

To evaluate the predictive models that have been defined throughout the study, a series of metrics commonly used in supervised machine learning applications were used [41,59]. The metrics used are as follows:(5)$$Accuracy=\frac{TP+TN}{TP+TN+FP+FN}$$
(6)$$Precision=\frac{TP}{TP+FP}$$
(7)$$Recall=\frac{TP}{TP+FN}$$
(8)$$F1=2\times \frac{Precision\times Recall}{Precision+Recall}$$
where *TP*, *FP*, *TN*, and *FN* are the true positive, false positive, true negative, and false negative cases, respectively. *Accuracy* is the ratio of the number of correct predictions to the total number of input samples, *precision* is the number of correct positive results divided by the number of positive results predicted by the classifier, *recall* is the true positive rate, and *F1* indicates the precision of the classifier (how many instances it classifies correctly) as well as its robustness (whether it misses a significant number of instances).

Once the areas with the most informative content were identified, the different ROIs were evaluated to select the exclusively necessary ones. The use of a greater number of analysis areas for the development of predictive models does not imply better results but can lead to data overfitting if the data are interdependent. On the contrary, it may happen that the number of ROIs used to develop the model is too small and that the model then fails to capture the general characteristics of the behavior under study [40]. The greater the number of features, the more noise and possible overfitting there may be to the data. The least number of features that manage to generalize the desired behavior should be selected for a correct performance.

The capacity of the identified ROIs for predictive models must be analyzed to determine those that best collect the behavior of the studied cases and generalize with new data. Different combinations of ROIs were analyzed. The process followed to define the ROIs considered in the final solution consisted of building different models of the proposed types, using the data from the dataset, and considering different ROIs in each case. According to the criteria and objectives set for this study to limit the computational cost of the algorithms developed, a reduced number of ROIs was initially considered. Specifically, only ROIs G and H, those with the greatest importance, were initially considered, and additional ROIs were progressively included in the ROI group, e.g., FGH, EFGH, DEFGH, and so on.

The results obtained in the analysis of the different possible combinations of ROIs are summarized in Figure 6. The assessment of each individual model was carried out by cross-validation with 10 stratified partitions [41,59]. This figure includes the average values obtained for different combinations of analysis ROIs, considering all types of models and all investigated features. Figure 6 shows that the average prediction capacity of the models increases if a greater number of regions is used. This is because, with a small number of ROIs, it is not possible to correctly predict all failure cases. As more features are considered, the models have more information to classify the cases. It may also be observed that not all areas provide the same information quality. Specifically, the combination DEFGH produces worse results than BCFGH. Moreover, there is a point in the process of adding ROIs to the training of the models where a higher number of ROIs hardly improves the results or even worsens them.

In addition to determining the areas of the board that are most representative of the operation mode, it is also necessary to determine which of the analyzed features certainly contain the information necessary to classify the different operating conditions. Table 1 includes the mean values of the analyzed metrics obtained for the different features, considering all combinations of ROIs and all predictive models analyzed in the study. The background color of each cell represents the scale of the particular value relative to all the values in the table. Higher values tend towards a greenish tone, lower values towards a reddish tone, and intermediate values are represented by yellowish tones between red and green depending on their value. The table clearly shows that the values of the approximation coefficients α for Grades 4 to 7 do not contain representative information, in the same way as the first component of PCA $\nu_{1}$, skewness $\mu_{3}$, and kurtosis $\mu_{4}$. The third approximation coefficient $\alpha_{2}$ and the third principal component $\nu_{3}$ are the features that provide the optimal classification results. On the other hand, the values calculated from the Fourier transform, both for phase analysis $F_{p}$ and for module $F_{m}$, achieve poor results.

Lastly, in order to correctly classify the different operating modes of the board, the particular capabilities offered by the different predictive models were analyzed. Table 2 shows the average metric results obtained for the different models analyzed and the combinations of ROIs considered. The values shown are average values obtained considering all features, and the criterion used for the background color of the cells is the same as in Table 1.

The predictive models were trained with the previously shown features and were experimentally configured in a heuristic way. The RF model was configured with 1000 estimators, with Gini criteria and with splits until the leafs contain two samples or less. The KNN was configured to group 15 neighbors and consider the Minkowski metric, and the ball-tree algorithm was used to perform the classification. The SVM model was configured with radial-basis kernel and one-vs-rest (ovr) decision function. The architecture used to configure the ANN was a sequential scheme with three hidden layers of 50, 300, and 40 neurons, each one with a ReLu activation function and a dropout of 0.4. The output layer contained nine neurons with softmax activation. The training was carried out for 80 epochs with a categorical cross-entropy loss function and an Adam optimizer with a learning rate of 0.01 (Figure 7a).

In most cases, the results were improved by providing more information to the model. The best results, from a global point of view, were obtained for the RF and ANN models. There is a range of ROIs for which the results are optimized for both of these models. In the case of RF, the best results were produced considering all ROIs identified. However, in the case of ANN, the best results were obtained by discarding ROI A from the total regions. This difference can be important to limit the computational cost and time required. The model that obtained the worst results was the SVM, probably because a suitable kernel had not been used, despite having tried the most common ones, such as linear, polynomial, and sigmoid kernels.

### 3.3. Inference Time Performance

According to the objectives of this study, while the prediction capacity is of vital importance, so are the computational cost, model development, and inference times. In the case of maintenance inspections, the time required to carry out an operation is a crucial factor.

The system proposed in this study is based on the analysis of preprocessed thermographic images, so the time consumption that must be taken into account first is that required to process the data collected by the IR sensor. These data processing techniques manage to extract the most relevant information from the complete data sequences, and the time involved in their application is in general short (less than 5 s). However, the time required for each process depends on its computational complexity and should be taken into account when assessing the suitability of a specific predictive system. In cases where time is not a limiting parameter, the precision of the predictions will be more highly weighted; on the other hand, in inspections with a short inspection time available, a balance must be reached between precision and execution times.

Table 3 shows the average times required to process the initial infrared data with the proposed algorithms. The criterion used for the background color of the cells in this table is the opposite to that in Table 1, since a better performance corresponds to lower time values. Thereby, greenish colors correspond to lower time values and represent better performance, while reddish colors correspond to higher time values and represent worse performance. It is observed that the slowest processing technique is TSR, and the fastest is the modulus of the Fourier transform.

It must be mentioned that the processing algorithms used in this assessment were configured in standard, similar conditions to have comparable results, but the times included in Table 3 can be reduced by using specific configurations with optimized parameter values. Only the processing algorithms were evaluated, ignoring the data loading and saving processes. The whole IR sequence was processed in every algorithm and all the PCA components and Fourier coefficients were calculated.

There are two other operations during the application of the predictive maintenance system that must be assessed in terms of computational cost. The first is the time required to train the predictive model, that is, the time it takes to configure the parameters of the model from the available data, in such a way that it is capable of making predictions with new unlabeled data. The other is the time that the model spends carrying out estimations or inferences on new data. In general, the inference time is less than the training time, although, as can be seen in Table 4, the KNN model performs the opposite effect. This table shows the average times required by the different models during their training and prediction stages. These times were estimated as the average amount of time needed to run the corresponding computations 20 times sequentially. The criterion used for the background color of the cells is the same as in Table 3. The Surface Book 2 platform (Microsoft Corp., Redmond, WA, USA) described in Section 2.4 was used to carried out these computations.

## 4. Discussion

In this section, a general procedure to define the optimal model for each specific maintenance application is proposed. This procedure considers the available dataset and the admissible operating times to justify the configuration accordingly through metric assessment.

According to the results presented in the previous section, the different proposed models produce different results. The optimal solution most likely depends on the final application for which the predictive system is intended. It is necessary to evaluate the results appropriately in order to obtain the most suitable predictive model for the maintenance of a particular piece of industrial equipment. The procedure followed to define the MANTRA predictive system, which can be generalized to any other predictive maintenance system based on thermographic measurements, is detailed in the following lines, schematically synthesized in Algorithm 1 and schematically represented in Figure 8.

First of all, a series of thermographic measurements of the device’s operation must be acquired. Conventional predictive machine learning systems require a relatively reduced dataset to be correctly configured compared to current deep learning models. Experience has demonstrated that a number of measures 10 times greater than the number of features used to train the model is usually necessary. In this study, this premise was fulfilled, analyzing a total of 180 cases for a maximum of eight features. In practice, the required number of measurements can be achieved quickly by performing regular inspections of the device of interest, so that the predictive model is successively readjusted over time and adapts to new anomalous operating conditions while increasing the accuracy of the predictions at the same time.

Once a minimum dataset is available, the regions or components of the device that provide the key information must be determined to classify the different operational situations correctly. For this, the thermal sequences acquired by the infrared camera must be processed with the appropriate techniques. In this study, several of the most commonly used algorithms in thermographic NDT applications were applied, and several of them demonstrated a high capacity to extract representative information of the actual operating modes from the sequences. However, this fact does not limit the application of other available processing techniques that can provide suitable results. The regions of the device to be monitored are determined in a preparatory stage by applying an analysis of importance with the processed thermal data. In this study, it is proposed to determine these informative regions using data analysis techniques based on decision trees and the Gini parameter. After identifying these regions, their location in maintenance inspections in real conditions is determined by automatically estimating the pose of the device by means of an RGB-D camera calibrated with an infrared sensor, enabling a determination of the component pose and its thermal evolution.

Subsequently, those regions that can distinguish different operating conditions must be selected to avoid overfitting. For this, the candidate models were built initially using a small number of the ROIs with the greatest importance, and the evaluation metrics were calculated for assessment. The feature employed in this assessment was the one used in the importance analysis to determine the candidate ROIs. The models were then rebuilt with the next-most informative ROI added to the previous set, and the metrics were recalculated. This process was repeated until the metrics did not improve the results when new ROIs were added, resulting in a final number of ROIs that obtained the highest metric values.

Next, the feature that best characterized the behavior of the inspected device was selected. For this purpose, the metrics of the candidate models were calculated for final ROI selection and for each of the features. Comparing the values obtained with the metrics, the feature that provided the best results was selected. Finally, to choose a predictive model that best meets the needs of the study case, the required training and prediction times of the different candidate models must be calculated. The final model was selected based on a multi-objective optimization problem, where prediction accuracy as well as data processing, training, and prediction times were assessed appropriately for the needs of real inspection.

In this study, as a result of the application of the proposed procedure, a predictive system based on an ANN model and using the third component $\nu_{3}$ of the principal component analysis measured on the BCFGH regions as the representative feature (ROIs defined in Figure 3b) was defined. The predictive model thus defined achieved an *accuracy* of 0.945, a *precision* of 0.951, a *recall* of 0.946, and an *F1* of 0.942, in cross-validation analysis. The average training time required by this model was approximately 1.6 s, considering processing and fitting, and the mean inference time after training was approximately 0.006 s. This option was selected because its accuracy is the highest, and the inference time is short. The time required for training is quite long compared to other models, but it is short enough to be carried out between maintenance tasks without affecting inspections.

Figure 7 shows the schematic architecture of the developed model and a confusion matrix obtained during the training stage. The model was evaluated with a 10-fold cross-validation strategy, so the confusion matrix shown in Figure 7b corresponds to a single specific fold. This case is one of the most unfavorable situations generated by the model, where it confused failures in Fuse 2 with an underload condition and confused a correct state with a failure in relay. These mispredictions can be justified by taking into account that the thermal responses of these conditions were very similar. The failure of the fuses caused their corresponding load to be zero, and the opposite load also experienced a decrease. Regarding the case of relay failure, it was a malfunction situation anticipating future failure, where the contact was reduced, but a current flow occurred; therefore, it resembled the reference state. The aforementioned values were reached after averaging the results obtained with all of the folds in the cross-validation process.

The model that obtained the lowest accuracy values was a SVM model using the kurtosis $\mu_{4}$ feature and the combination of ROIs BCFGH. It produced values of *accuracy* of 0.269, *precision* of 0.224, *recall* of 0.272, and *F1* of 0.203. This specific SVM model was not capable of identifying the existing non-linearity in the behavior of the board and failed in generalizing. However, all the SVM models required very short times for training and inference, around 0.003 s and 0.0007 s, respectively, and obtained an *accuracy* of up to 0.795 using the third polynomial coefficient $\alpha_{2}$ feature applied on the ROIs DEFGH. The model with the second highest accuracy was the RF model with an *accuracy* of 0.897, a *precision* and *recall* of 0.904, and an *F1* of 0.881. The main drawback of the RF models was the time required for inference, which was around 1.5 s in all cases, that could limit the inspection capability in real industrial conditions. The optimum predictions obtained with the KNN model produced an *accuracy* of 0.795, a *precision* of 0.797, a *recall* of 0.798, and an *F1* of 0.776. This KNN model was applied on the regions DEFGH using the third polynomial coefficient $\alpha_{2}$ feature. The KNN models required the shortest time for training, around 0.001 s in all the analyzed cases, but the second maximum time for inference, around 0.12 s.
**Algorithm 1** An algorithm proposed to develop a robust and fast predictive maintenance system based on thermographic images1:Select industrial equipment2:Generate thermographic labeled dataset (reference + abnormal conditions)3:process_time ← thermographic data processing4:ROI candidates ← importance analysis**Ensure:** 
Sort ROIs in descending order of importance5:**while** metric((N+1)${}_{ROIs}$) < metric(N${}_{ROIs}$)**do**6:       $N_{ROIs}\leftarrow$
(N+1)${}_{ROIs}$7:**end while**8:**for** feature in features **do**9:       metric(feature)10:**end for**11:feature← max(metric(feature))12:**for** model in models **do**13:       train_time(model)14:       prediction_time(model)15:       metric(model)16:**end for**17:model← optimize(process_time, train_time, prediction_time, metric)

## 5. Conclusions

The maintenance of industrial equipment, machines, or facilities tends to extend its useful life, to obtain correct performance for a longer time, to improve its efficiency, to reduce the number of failures, and to increase the safety of its use. Maintenance management has evolved significantly over time. Progress has been made in the way maintenance is applied to industrial equipment.

This study provides advances towards the robust application of the ML combined with IRT to maintenance applications of industrial equipment. A new module for the MANTRA system [28] that informs about failures or abnormal behaviors of industrial equipment analyzing thermal images was developed. Moreover, AR was also combined with the latter technologies and implemented in a portable device making the MANTRA system, which is, to our knowledge, the first AR system that has been used in combination with IRT in the field of industrial maintenance. This MANTRA system enabled quick hand-held thermal inspections using information of the previously calculated object pose. Experimental data were produced in this study by monitoring electric industrial equipment and inducing malfunctions in a controlled way. Different machine learning strategies were analyzed and assessed to finally define the optimal procedure to be implemented in the MANTRA system. The accuracy obtained by the optimal algorithm was up to 94% at an inference time of 0.006 s.

Future studies will try to validate the methodology developed to define the predictive inspection system by applying it to other industrial equipment, not only electrical but also mechanical.

## Figures and Tables

**Figure 1 sensors-22-00613-f001:**
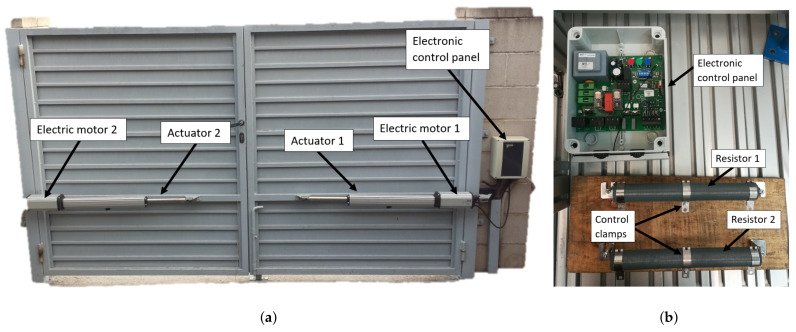
The study case used to develop and assess the predictive module: (**a**) configuration in real conditions; (**b**) simulated configuration at the laboratory level.

**Figure 2 sensors-22-00613-f002:**
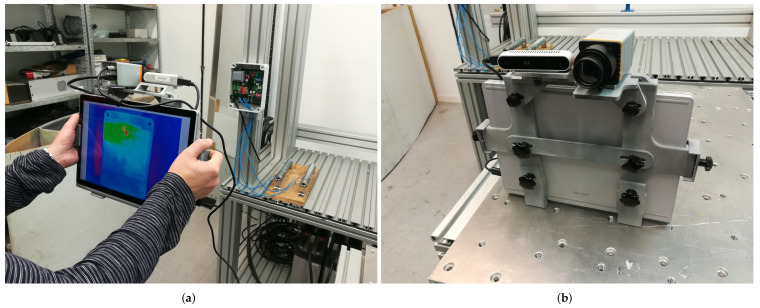
Sensors included in the MANTRA system: (**a**) calibration of the sensors with the electronic board; (**b**) IR and RGB-D cameras mounted in the MANTRA demonstrator.

**Figure 3 sensors-22-00613-f003:**
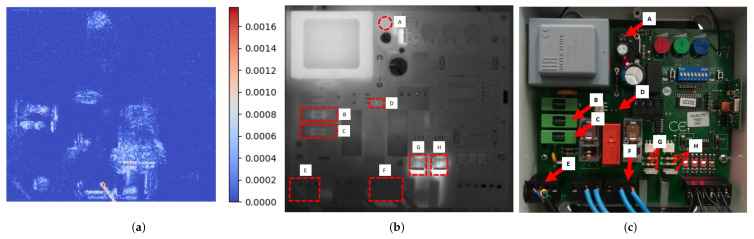
Identification of the regions of interest: (**a**) results of importance analysis; (**b**) location of the most informative regions; (**c**) identification of the corresponding electronic elements.

**Figure 4 sensors-22-00613-f004:**
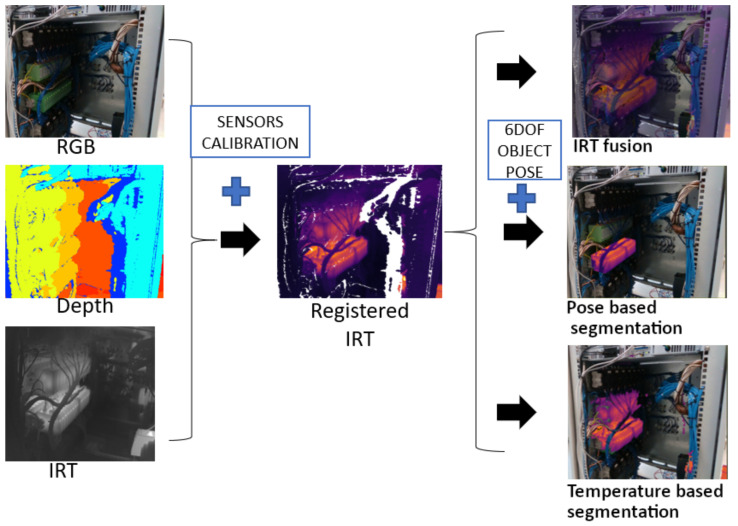
Schematic of the MANTRA system for IRT segmentation based on a temperature range or on a 6DOF object pose of an industrial computer.

**Figure 5 sensors-22-00613-f005:**
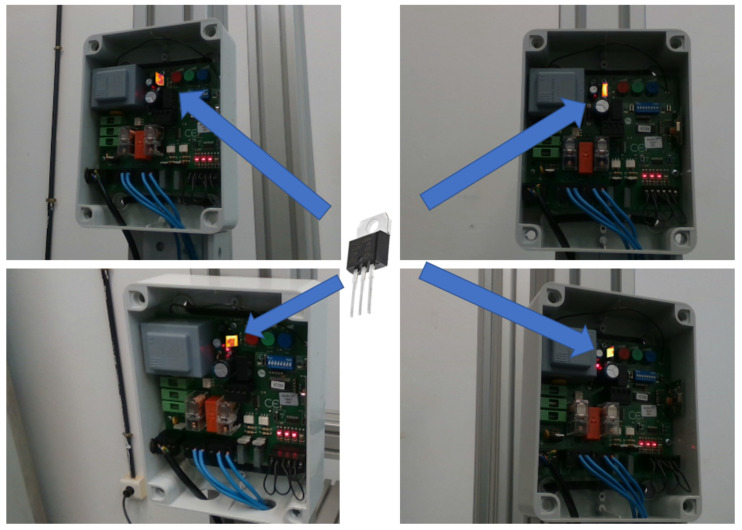
Automatic IRT thermal segmentation of the triac component controlling the power source of the electronic board from different points of view.

**Figure 6 sensors-22-00613-f006:**
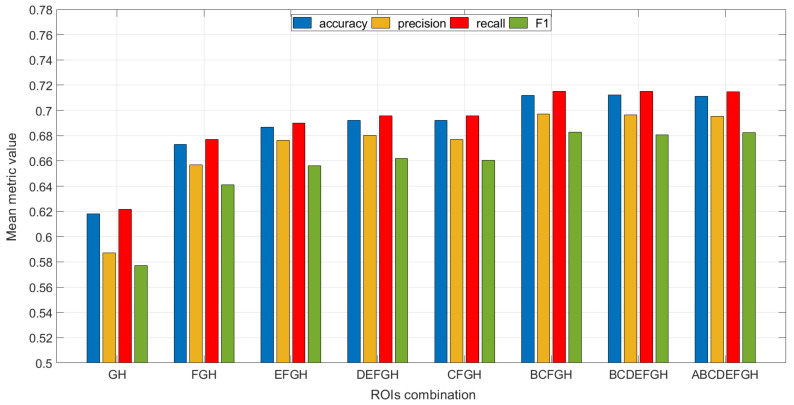
Mean values of the selected metrics for different ROI groups.

**Figure 7 sensors-22-00613-f007:**
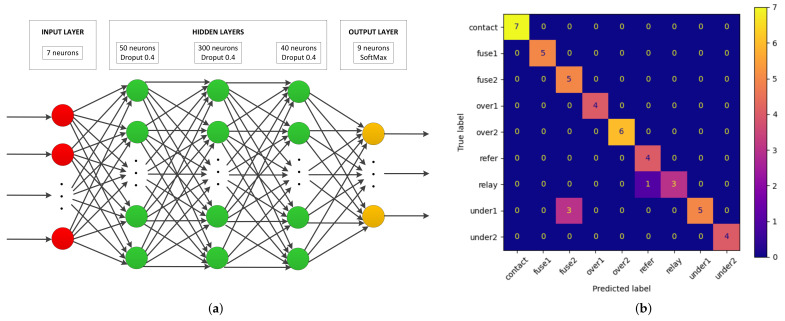
Predictive model implemented in the MANTRA system: (**a**) architecture of the ANN; (**b**) confusion matrix obtained with a fold of the training dataset.

**Figure 8 sensors-22-00613-f008:**
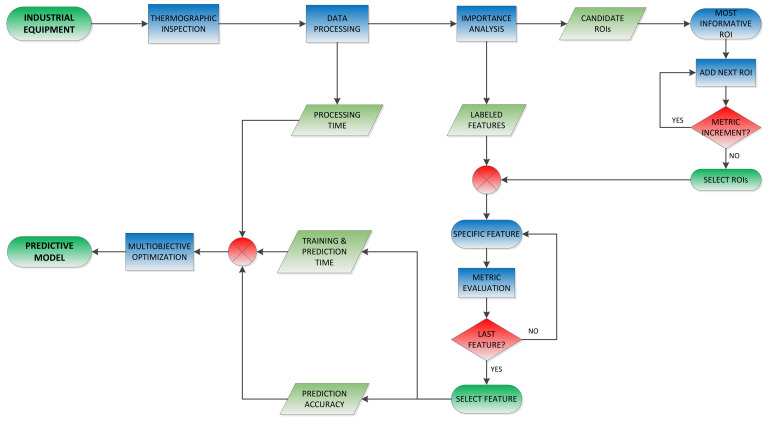
Flowchart of the proposed procedure to develop a robust and fast predictive maintenance system based on thermographic images.

**Table 1 sensors-22-00613-t001:** Values of the selected metrics obtained for the analyzed features.

	$\alpha_{0}$	$\alpha_{1}$	$\alpha_{2}$	$\alpha_{3}$	$\alpha_{4}$	$\alpha_{5}$	$\alpha_{6}$	$\alpha_{7}$
*accuracy*	0.742	0.767	0.794	0.663	0.605	0.608	0.588	0.612
*precision*	0.725	0.778	0.791	0.650	0.566	0.583	0.559	0.588
*recall*	0.749	0.766	0.796	0.667	0.607	0.612	0.594	0.619
*F1*	0.714	0.745	0.770	0.632	0.556	0.575	0.548	0.578
	$F_{m}$	$F_{p}$	$\nu_{1}$	$\nu_{2}$	$\nu_{3}$	$\nu_{4}$	$\mu_{3}$	$\mu_{4}$
*accuracy*	0.679	0.667	0.589	0.730	0.749	0.719	0.557	0.593
*precision*	0.646	0.673	0.518	0.711	0.756	0.698	0.536	0.567
*recall*	0.687	0.672	0.587	0.733	0.753	0.719	0.560	0.594
*F1*	0.642	0.641	0.528	0.706	0.736	0.689	0.512	0.546

**Table 2 sensors-22-00613-t002:** Values of the selected metrics obtained for the analyzed models.

		GH	F-H	E-H	D-H	CF-H	BCF-H	B-H	A-H
KNN	*accuracy*	0.638	0.679	0.684	0.674	0.683	0.696	0.686	0.684
	*precision*	0.627	0.673	0.686	0.669	0.674	0.689	0.674	0.672
	*recall*	0.642	0.684	0.689	0.678	0.688	0.701	0.690	0.688
	*F1*	0.608	0.652	0.661	0.648	0.656	0.672	0.658	0.656
RF	*accuracy*	0.632	0.693	0.713	0.725	0.715	0.733	0.736	0.737
	*precision*	0.631	0.699	0.726	0.732	0.722	0.738	0.736	0.739
	*recall*	0.633	0.696	0.716	0.728	0.717	0.736	0.738	0.741
	*F1*	0.611	0.678	0.697	0.707	0.696	0.717	0.711	0.722
SVM	*accuracy*	0.580	0.624	0.629	0.639	0.641	0.663	0.656	0.656
	*precision*	0.517	0.588	0.596	0.609	0.606	0.629	0.620	0.619
	*recall*	0.584	0.627	0.632	0.642	0.644	0.665	0.658	0.658
	*F1*	0.520	0.577	0.584	0.596	0.597	0.621	0.614	0.614
ANN	*accuracy*	0.621	0.694	0.718	0.728	0.727	0.753	0.768	0.766
	*precision*	0.571	0.667	0.695	0.709	0.704	0.731	0.754	0.749
	*recall*	0.626	0.699	0.722	0.732	0.732	0.757	0.771	0.770
	*F1*	0.568	0.655	0.681	0.694	0.691	0.721	0.738	0.736

**Table 3 sensors-22-00613-t003:** Values expressed in seconds of the time spent by the thermal data processing techniques.

N${}^{\circ }$ ROIs	α	$F_{p}$	$F_{m}$	ν	$\mu_{3}$	$\mu_{4}$
2	0.0283	0.0016	0.0013	0.0017	0.0147	0.0143
3	0.0430	0.0023	0.0018	0.0021	0.0218	0.0208
4	0.0576	0.0031	0.0025	0.0025	0.0286	0.0286
5	0.0719	0.0037	0.0031	0.0027	0.0361	0.0355
6	0.0854	0.0045	0.0036	0.0028	0.0427	0.0423
7	0.1007	0.0052	0.0042	0.0030	0.0499	0.0495
8	0.1154	0.0062	0.0051	0.0031	0.0568	0.0563

**Table 4 sensors-22-00613-t004:** Values of the time spent by the analyzed predictive models during the training and prediction stages, and considering different number of ROIs, expressed in seconds.

	TRAINING				PREDICTION			
**N${}^{\circ }$ ROIs**	**KNN**	**RF**	**SVM**	**ANN**	**KNN**	**RF**	**SVM**	**ANN**
2	0.00093	1.2034	0.00211	1.6104	0.1188	0.1341	0.00065	0.00635
3	0.00094	1.2043	0.00246	1.5987	0.1108	0.1392	0.00067	0.00620
4	0.00096	1.1942	0.00236	1.6531	0.1191	0.1455	0.00068	0.00602
5	0.00099	1.1979	0.00265	1.6506	0.1210	0.1459	0.00069	0.00620
6	0.00101	1.1974	0.00261	1.6168	0.1191	0.1435	0.00068	0.00601
7	0.00102	1.1999	0.00259	1.6083	0.1191	0.1497	0.00069	0.00605
8	0.00103	1.1822	0.00261	1.6057	0.1197	0.1535	0.00073	0.00622

## Data Availability

Not applicable.

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
