# Peer review of "Towards the Automation of Infrared Thermography Inspections for Industrial Maintenance Applications"

_sensors, 2022, doi:10.3390/s22020613_

Round 1
Reviewer 1 Report
The article shows a well-described and articulated case study.
Some suggestions.
- Before the description of the TSR add a descriptive part of the thermography, in general
- All the algorithms are here described as known in the literature. Maybe it is better to have a section for the theory, in general and a separate section, material and methods (maybe after the case study) to describe the use of these algorithms for this work (example: number of frames, polynomial degree for TSR and so on)
- For the different tables pages 13-16 precise the legend for the different colors
- Use a unique legend for the ROIS; they are A, B, C...or 2, 3, 4 as reported in table 4?
- Could be interesting to precise the ROI dimension
- Replace algorithm 1 with a flowchart
- Did you perform PCA or SVD algorithm? SVD is generally very fast. The times indicated in table 3 depend on many factors. For example, what has been saved for the TSR? If only the coefficients are used you can save only those, without asking for the polynomial.
- Paragraph 2.4 it is necessary to specify an mm/pixel ratio for the two cameras
Author Response
Authors would like to thank the reviewer for the valuable review and suggestions. The responses to the comments are explained below. Additionally, a version of the revised manuscript with the changes marked is available in the attachment link (Please see the attachment).
- Before the description of the TSR add a descriptive part of the thermography, in general
A brief general description of the thermography has been added at the beginning of section 2.1. Thermographic Selected Features, together with some new citations.
- All the algorithms are here described as known in the literature. Maybe it is better to have a section for the theory, in general and a separate section, material and methods (maybe after the case study) to describe the use of these algorithms for this work (example: number of frames, polynomial degree for TSR and so on)
We really appreciate this recommendation. It makes sense to have the theory before in the paper and include the specific configurations afterwards. We have made several tests on this sense, with different organization of the contents, but the structure was unbalanced in all the cases. We humbly consider the current organization makes better paper structure.
- For the different tables pages 13-16 precise the legend for the different colors.
Several sentences have been included next to the references to these tables, explaining the background color criteria used in each case.
- Use a unique legend for the ROIS; they are A, B, C...or 2, 3, 4 as reported in table 4?
Table 4 includes values of the time required by the models considering different number of regions, independently of which ROIs they specifically are. A sentence has been included in the caption of this table to highlight this condition.
- Could be interesting to precise the ROI dimension.
The dimensions of the ROIs have been indicated, together with the electronic components to which they correspond, in section 3.1. Selection of Regions of Interest.
- Replace algorithm 1 with a flowchart.
We have made a flowchart and included it in section 4. Discussion.
Authors agree that a flowchart is a suitable graphic resource for readers to understand the procedure proposed to develop a predictive maintenance system. Since this procedure is a fundamental part of the results obtained in the study, we consider its understanding is of the utmost importance. Therefore, we would like to keep both the algorithm and the flowchart in the paper to complement each other and provide clarification to readers.
- Did you perform PCA or SVD algorithm? SVD is generally very fast. The times indicated in table 3 depend on many factors. For example, what has been saved for the TSR? If only the coefficients are used you can save only those, without asking for the polynomial.
We applied PCA using SVD algorithm. Only the coefficients were saved after TSR, but the time dedicated to the data saving process was not taken into account in Table 3.
Authors completely agree that the times indicated in Table 3 depend on many factors. Standard configuration and similar conditions were considered in all the algorithms to have comparable results, but the times included in Table 3 can be reduced by using specific configurations with optimized parameter values.
A short note about this issue has been included in the text near Table 3 to clarify it.
- Paragraph 2.4 it is necessary to specify an mm/pixel ratio for the two cameras
The values of mm/pixel corresponding to both cameras have been included in the paper in Paragraph 2.4. Hardware and Software Equipment.
Kind regards.

Reviewer 2 Report
The authors presented their work on the use of ML for automation of IRT inspection for industrial maintenance. They evaluated 4 processing algorithms and 4 supervised ML models in this work.
I believe this will be of interest to the readers and would like to suggest the following for further improvement of the article:
- Figure 3 listed regions "A" to "H" but did not label/describe what they are, e.g. contact, fuse, relay, etc. The regions should be labelled/described in the article since the confusion matrix on Figure 7(b) mentioned these regions.
- For Figure 7(b), the label for the x-axis is missing. I believe it should be "Predicted Label"?
- Line 486 mentioned Ff and Fp, however Table 1 mentioned Fm and Fp. I believe there is a typo mistake?
- Line 637 mentioned that "optimal algorithm was up to 94% at an inference time of 0.006s". I did not see an elaborated discussion of that in the article. Can the author please elaborate/highlight that explicitly in the article? This will be useful to readers who plan to implement the authors' suggestion.
Author Response
Authors would like to thank the reviewer for the valuable review and suggestions. The responses to the comments are explained below. Additionally, a version of the revised manuscript with the changes marked is available in the attachment link (Please see the attachment).
- Figure 3 listed regions "A" to "H" but did not label/describe what they are, e.g. contact, fuse, relay, etc. The regions should be labelled/described in the article since the confusion matrix on Figure 7(b) mentioned these regions.
This is an important error in the manuscript. The description was included in a previous revision of the paper but it was accidentally removed during the writing process. We really appreciate this comment.
A brief description of the electronic components that each region corresponds to has been included in section 3.1 Selection of ROIs of interest.
- For Figure 7(b), the label for the x-axis is missing. I believe it should be "Predicted Label"?
The label for x axis has been made visible.
- Line 486 mentioned Ff and Fp, however Table 1 mentioned Fm and Fp. I believe there is a typo mistake?
That is correct. It was a typo and has been amended in the revised paper.
- Line 637 mentioned that "optimal algorithm was up to 94% at an inference time of 0.006s". I did not see an elaborated discussion of that in the article. Can the author please elaborate/highlight that explicitly in the article? This will be useful to readers who plan to implement the authors' suggestion.
The discussion has been completed in the last paragraph of section 4. Discussion. The most remarkable results obtained with the other models have been discussed.
Kind Regards.

Reviewer 3 Report
As the authors cannot expect the readers are experts in AI-procedures named by numerously applied terminology without specific citation. So check whether to enhance the number of citations.
Author Response
Authors would like to thank the valuable comment. Authors completely agree with the reviewer that AI terminology could be overwhelming at first sight. To try to mitigate this issue, new citations have been added within the text to provide references for specific concepts and terminology. Specifically:
- JIANG, Hui. Machine Learning Fundamentals: A Concise Introduction. Cambridge University Press, 2021.
- HACKELING, Gavin. Mastering Machine Learning with scikit-learn. Packt Publishing Ltd, 2017.
- KETKAR, Nikhil. Machine Learning Fundamentals. In Deep Learning with Python. Apress, Berkeley, CA, 2017. p. 7-16.
A version of the revised manuscript with the changes marked is available in the attachment link (Please see the attachment).
Kind Regards.
